# The Role of Immunohistochemistry Markers in Endometrial Cancer with Mismatch Repair Deficiency: A Systematic Review

**DOI:** 10.3390/cancers14153783

**Published:** 2022-08-03

**Authors:** Amelia Favier, Justine Varinot, Catherine Uzan, Alex Duval, Isabelle Brocheriou, Geoffroy Canlorbe

**Affiliations:** 1Department of Gynecological and Breast Surgery and Oncology, Pitié-Salpêtrière, Assistance Publique des Hôpitaux de Paris (AP-HP), University Hospital, 75013 Paris, France; catherine.uzan@aphp.fr (C.U.); geoffroy.canlorbe@aphp.fr (G.C.); 2Centre de Recherche Saint-Antoine, Equipe Instabilité des Microsatellites et Cancer, Equipe Labellisée par la Ligue Nationale Contre le Cancer, Unité Mixte de Recherche Scientifique 938 and SIRIC CURAMUS, INSERM, Sorbonne Université, 75012 Paris, France; alex.duval@inserm.fr; 3Department of Pathology, Tenon Hospital, HUEP, Sorbonne Université, 75020 Paris, France; justine.varinot@aphp.fr; 4Centre de Recherche Saint-Antoine (CRSA), INSERM UMR_S_938, Cancer Biology and Therapeutics, Sorbonne University, 75012 Paris, France; 5Department of Pathology, Pitié-Salpêtrière Hospital, Sorbonne University, 75013 Paris, France; isabelle.brocheriou@aphp.fr

**Keywords:** endometrial cancer, mismatch repair-deficient, immunohistochemistry markers, microsatellite instability, diagnosis, prognosis

## Abstract

**Simple Summary:**

Identification of mismatch repair-deficient tumors (MMRd), which occur in up to 30% of all endometrial cancers (EC), has become unavoidable for therapeutic management, clinical decision making, and prognosis. The objective of this systematic review was to summarize our current knowledge of the role of immunohistochemistry (IHC) markers for identifying MMRd tumors in EC. IHC with the expression of four proteins and MLH1 promoter methylation remains the reference of choice for diagnosis because it is reproducible and applicable in routine clinical practice. Further studies are needed to evaluate IHC in comparison with molecular tests including artificial intelligence, in terms of both efficacy and medical/economic aspects.

**Abstract:**

The objective of this systematic review was to summarize our current knowledge of the role of immunohistochemistry (IHC) markers for identifying mismatch repair-deficient (MMRd) tumors in endometrial cancer (EC). Identification of MMRd tumors, which occur in 13% to 30% of all ECs, has become critical for patients with colorectal and endometrial cancer for therapeutic management, clinical decision making, and prognosis. This review was conducted by two authors applying the Preferred Reporting Items for Systematic Reviews and Meta-Analyses (PRISMA) guidelines using the following terms: “immunohistochemistry and microsatellite instability endometrial cancer” or “immunohistochemistry and mismatch repair endometrial cancer” or “immunohistochemistry and mismatch repair deficient endometrial cancer”. Among 596 retrieved studies, 161 fulfilled the inclusion criteria. Articles were classified and presented according to their interest for the diagnosis, prognosis, and theragnostics for patients with MMRd EC. We identified 10, 18, and 96 articles using IHC expression of two, three, or four proteins of the MMR system (MLH1, MSH2, MHS6, and PMS2), respectively. MLH1 promoter methylation was analyzed in 57 articles. Thirty-four articles classified MMRd tumors with IHC markers according to their prognosis in terms of recurrence-free survival (RFS), overall survival (OS), stage, grade, and lymph node invasion. Theragnostics were studied in eight articles underlying the important concentration of PD-L1 in MMRd EC. Even though the role of IHC has been challenged, it represents the most common, robust, and cheapest method for diagnosing MMRd tumors in EC and is a valuable tool for exploring novel biotherapies and treatment modalities.

## 1. Introduction

In 2020, an estimated 417,367 new cases of endometrial cancer (EC) were diagnosed worldwide, and 97,370 women are estimated to have died from the disease [1]. Most ECs (90%) are diagnosed at an early stage with a 5 year survival rate of 95% [2]. However, once the cancer spreads to distant organs, the prognosis is poor with a 5 year survival rate of 17% [3].

In the past decade, numerous studies have explored prognostic factors such as pathologic type, histologic grade, lymphovascular involvement, and tumor staging but with insufficient reproducibility. Investigation has, therefore, turned to gene carcinogenesis such as molecular alterations to provide a new prognostic classification.

In 2017, the Proactive Molecular Risk Classifier for Endometrial Cancer (ProMisE) described four molecular prognostic groups: ultramutated DNA polymerase epsilon (POLE) tumors which have the best prognosis, microsatellite instability (MSI) or mismatch repair (MMR)-deficient (MMRd) hypermutated tumors with intermediate prognosis, p53 abnormal tumors with the worst prognosis, and tumors with copy-number low alterations with good to intermediate prognosis [4]. This classification is consistent with The Cancer Genome Atlas (TCGA) and applicable in clinical practice [5,6].

The MMRd tumor phenotype represents 17–33% of all ECs [7]. Originally, this molecular phenotype was found in patients with germinal mutations known as Lynch syndrome (LS) [8], which conveys a lifetime risk of EC of 60% [9]. Furthermore, this specific molecular genetic alteration, resulting from a defect in the MMR genes hMLH1, hMSH2, hMSH6, or hPMS2 is found in sporadic tumors called MMRd or Lynch-like syndrome tumors [10]. Tumors without defect in MMR genes are called mismatch repair-proficient (MMRp). The accumulation of insertions or deletions of nucleotides into coding repeat sequences results in an increase in lymphocyte infiltration, and the phenotype is, therefore, a possible candidate for immunotherapy [11]. Thus, identification of MMRd tumors has become critical for patients with EC for therapeutic management, clinical decision making, and prognosis.

In addition to MMR testing and to better identify risk groups currently included in the latest European Society of Gynecological Oncology/European Society for Radiotherapy and Oncology/European Society of Pathology (ESGO/ESTRO/ESP) [12], IHC markers such as p53 or POL-E have been proposed in a diagnostic algorithm [13]. Recent studies showed that abnormal p53 IHC reliably identifies cases with TP53 mutation (representing 25% of all EC) in EC biopsies (94% specificity and 91% sensitivity) [14]. POL-E mutations representing 8.59% of all ECs are mainly presented at earlier stages I–II (89.51%) and at the highest grade III (51.53%) [15]. Other IHC markers have been studied such as estrogen receptor, progesterone receptor, HER-2, and Ki67; however, no single marker was found to be indicative of EC often enough to allow routine use in the subclassification of EC [16].

Since 2018, the National Comprehensive Cancer Network guidelines recommend universal testing of all ECs for MSI/MMRd tumors [17]. Many oncologic centers use immunohistochemistry (IHC) for such testing as it is cheap and of high sensitivity, specificity, and reproducibility [5,18]. The European Society for Medical Oncology (ESMO) guidelines recommend the MMR-IHC test [12] for all patients with EC irrespective of histologic subtype. Molecular tests, such as PCR-based molecular testing using five of the eight mononucleotide or dinucleotide repeats (BAT-25, BAT-26, NR-21, NR-24, NR-27, D5S346, D2S123, and D17S250) or next-generation sequencing (NGS) can be used as an alternative or when IHC is indeterminate. Nevertheless, it is unclear which technique is recommended, reliable, and suitable for use in routine practice.

To the best of our knowledge, there is no recent review of the value of IHC markers to identify MMRd EC phenotypes or LS-related EC. The objective of this systematic review was, therefore, to summarize our current knowledge of the role of IHC markers in MMRd EC focusing on prognosis, diagnosis, and theragnostics.

## 2. Materials and Methods

This systematic review was carried out in accordance with the Preferred Reporting Items for Systematic Reviews and Meta-Analyses (PRISMA) guidelines using the following databases: MEDLINE, PubMed (the Internet portal of the National Library of Medicine, http://www.ncbi.nlm.nih.gov/sites/entrez?db=pubmed; accessed on 30 September 2021), the Cochrane Library, Cochrane databases “Cochrane Reviews”, and “Clinical Trials” (http://www3.interscience.wiley.com/cgi-bin/mrwhome/106568753/HOMEDARE; accessed on 30 September 2021). The PRISMA checklist is provide in the Appendix A. This systematic review has not been registered in PROSPERO.

Two independent reviewers (A.F. and G.C.) retrieved information about articles such as authors, date of publication, journal, study design, population characteristics, type of screening test, number of MMRd tumors studied, outcomes, and survival rates. Any discrepancies about the eligibility of an article were discussed between the two reviewers, with the senior author (G.C.) having the final word.

The search was conducted with the following terms: “immunohistochemistry and microsatellite instability endometrial cancer” or “immunohistochemistry and mismatch repair endometrial cancer” or “immunohistochemistry and mismatch repair deficient endometrial cancer”. Lynch syndrome patients were also included.

The database search was further supplemented with original articles, reviews, and meta-analyses. All duplicates were removed. Only articles published in English between 1 July 1999 and 30 September 2021 were included. Among 596 retrieved studies, 161 fulfilled the inclusion criteria. The exclusion criteria were no detail of IHC, no MMRd tumors, no detail of EC, or MMRd diagnosed with molecular approach only.

The articles were classified and presented according to the use of IHC for the diagnosis, prognosis, and theragnostics for patients with MMRd endometrial tumors. We summarized the results of the main outcomes, as it was not feasible to pool findings due to the heterogeneity among the studies in terms of patients, tumor characteristics, technique used, statistical analysis, and outcome measures.

## 3. Results

The PRISMA flow diagram is presented in Figure 1.

This systematic review identified 157 original articles and four literature reviews reporting the role of IHC markers in MMRd EC. Of the 155 original articles studying neoplastic endometrial tissue, the average number of endometrial tumor samples was 389 (minimum four, maximum 5917). A total of 103 articles reported analysis from formalin-fixed paraffin-embedded (FFPE) tissue, along with eight from FFPE and frozen tissue, 10 from frozen tissue, and one from fresh tissue.

### 3.1. IHC in the Diagnosis of MMRd Tumors in EC

IHC expression of two proteins of the MMR system in EC is presented in Table 1.

We identified 10 articles (published between 1999 and 2020) using IHC expression of two proteins of the MMR system in EC, nine of which reported the loss of the MLH1 and MSH2 proteins, and one of which reported the loss of MSH6 and PMS2 [19,20,21,22,23,24,25,26,27,28]. Of the 10 studies, nine used FFPE tissue and two used frozen tissue, including one article comparing neoplastic endometrial tissue with the surrounding healthy tissue. A germline mutation was screened in four articles [21,22,25].

The average number of MMRd tumors included was 17.9 (minimum six, maximum 53). The percentage of MMRd tumors in the cohorts ranged from 13.5 to 100%. The loss of the MLH1 and MSH2 proteins in the MMRd tumors ranged from 12 to 83.3% and from 1.1 to 86%, respectively. The loss of the MSH6 and PMS2 proteins was reported at 0% and 100%, respectively. One article reported MLH1 promoter methylation in 77% of the MMRd tumors.

IHC expression of three proteins of the MMR system in EC is presented in Table 2.

We identified 18 articles (published between 2002 and 2020) using IHC expression of three proteins of the MMR system (MLH1, MSH2, and MSH6) in EC, of which 15 used FFPE tissues, and three used frozen tissue [29,30,31,32,33,34,35,36,37,38,39,40,41,42,43,44,45,46]. Two articles did not describe the technique used, and five articles compared neoplastic endometrial tissue with the surrounding healthy tissue.

A germline mutation was screened in 11 of the articles [29,31,33,34,35,36,39,40,42,44,46]. The average number of MMRd tumors included was 34.6 (minimum six, maximum 164). The percentage of MMRd tumors in the cohorts ranged from 20% to 100%. The loss of the MSH2, MLH1, and MSH6 proteins in the MMRd tumors ranged from 0% to 55.2%, from 0% to 100%, and from 0% to 66.7%, respectively. Four articles reported MLH1 promoter methylation in 14% to 70% of the MMRd tumors.

IHC expression of four proteins of the MMR system in EC is presented in Table 3.

We identified 96 articles (published between 2008 and 2021) using IHC expression of four proteins of the MMR system (MLH1, MSH2, MSH6, and PMS2) in EC, of which 73 used FFPE tissues, three used frozen tissue, and one used tumor cells [40,47,48,49,50,51,52,53,54,55,56,57,58,59,60,61,62,63,64,65,66,67,68,69,70,71,72,73,74,75,76,78,79,80,81,82,83,84,85,86,87,88,89,90,91,92,93,94,95,96,97,98,99,100,101,102,103,104,105,106,107,108,109,110,111,112,113,114,115,116,117,118,119,120,121,122,123,124,125,126,127,128,129,130,131,132,133,134,135,136,137,138,139,140,141,142]. Twenty-one articles did not describe the technique, and eight compared neoplastic endometrial tissue with surrounding healthy tissue.

A germline mutation was screened in 53 of the articles [49,50,51,53,54,56,57,61,62,63,64,68,69,70,71,72,74,75,79,82,83,84,85,86,87,88,89,90,92,93,94,96,97,104,106,110,112,113,114,116,119,121,122,123,124,127,128,129,133,136,140]. The average number of MMRd tumors included was 98.5 (minimum one, maximum 2563). The percentage of MMRd tumors in the cohorts ranged from 5% to 100%. The loss of the MSH2, MLH1, MSH6, and PMS2 proteins in the MMRd tumors ranged from 0% to 75%, from 0% to 100%, from 0% to 80%, and from 2% to 100%, respectively. Twenty-seven articles reported MLH1 promoter methylation in 14% to 100% of the MMRd tumors.

MLH1 promoter methylation analysis in MMRd EC is presented in Table 4.

We identified 57 articles analyzing MLH1 promoter methylation [19,26,32,33,34,36,40,42,46,47,52,55,57,58,59,69,71,75,79,81,82,83,84,85,86,90,91,94,95,96,99,100,112,113,116,122,124,125,127,128,129,136,137,139,140,141,142,143,144,145,146,147,148,149,150,151,152]. Among these, 39 articles used bisulfite conversion of tumor DNA with PCR amplification, five used a methylation-specific multiplex ligation dependent probe-amplification technique, two used NGS, and one used a pyrosequencing technique. The MLH1 promoter methylation gene was found in 14% to 100% of the MMRd tumors. It was tested in order to identify a germline or somatic mutation.

### 3.2. IHC in the Prognosis of MMRd Tumors in EC

We identified 34 articles studying the prognosis of patients with MMRd tumors in EC [25,26,51,54,60,63,70,72,75,76,84,89,92,99,102,105,106,115,120,126,130,131,132,133,135,140,141,153,154,155,156,157,158,159]. The studies included an average of 191.5 samples of MMRd tumors (minimum 12, maximum 892).

#### 3.2.1. Survival

The recurrence-free survival (RFS) and overall survival (OS) of patients with MMRd tumors in EC compared with patients with MMRp tumors are presented in Table 5.

MMRd tumors were associated with better RFS than MMRp tumors in four articles [75,99,102,158]. One reported a recurrence of 10% in patients with MMRd tumors versus 42% in those with MMRp tumors [99] with a median follow-up of 31 months for both MMR groups. Another reported a hazard ratio (HR) of 0.61 [102]. A third one reported a 5 year RFS of 95% or 87.9% versus 80.4% for MMRp tumors, 93% for POLE tumors, 52% for no specific molecular profile tumors, and 42% for p53 tumors [75]. The last study reported 36 months of RFS versus 9 months for MMRp tumors [158]. Two articles reported a worse RFS for patients with MMRd tumors than for those with MMRp tumors [54,120]; the 5-year RFS for MMRd tumors was 66% in the first and 71.1% in the second compared with 89% and 97.6% for MMRp tumors.

MMRd tumors were associated with better OS than MMRp tumors in two articles [99,102]. With a median follow-up of 31 months, the mortality rate of patients with MMRd tumors was 13.1% compared to 36.1% for MMRp tumors (HR: 0.80). Two articles found a worse OS for patients with MMRd tumors with a 5 year OS of 74% and 71% compared with 86% and 100% for MMRp tumors [54,120]. Two articles showed an intermediate prognosis for MMRd tumors [126,131]. The 5 year RFS and OS for MMRd tumors were 71.4% and 81.3% compared with 98% and 98% for POLE tumors and 48% and 54% for p53 tumors, respectively. Lastly, in nine articles, there were no significant differences in the RFS and OS or clinical features between patients with MMRd or MMRp tumors [60,63,76,133,140,141,153,154,155].

#### 3.2.2. Pathologic Characteristics

MMRd tumors were associated with a lower grade EC in three articles [106,115,156]; 83% to 90% of low-grade EC tumors had the MMRd phenotype compared with 31.4% to 77% of the MMRp tumor phenotypes. Five articles showed that MMRd tumors were associated with a higher-grade EC [25,70,130,132,157]; 33% of higher-grade tumors were found to be MMRd and associated with endometrioid and mixed histologic tumors.

MMRd tumors were associated with a lower FIGO stage in three articles [92,106,115]; 58% to 90% of low-stage EC were MMRd compared with 41.8% to 78% for MMRp tumors. MMRd tumors were associated with higher FIGO stage in six articles [25,72,132,135,157,160]; 58% to 66.7% in higher-stage EC were MMRd compared with 17.6% to 40.6% for MMRp tumors.

MMRd tumors were associated with less lymph node invasion in one article [157], with 44.6% compared with 55.4% in MMRp tumors.

MMRd tumors were associated with greater lymphovascular invasion in one article [92], with 58% for MMRd tumors and 48% for MMRp tumors.

In three articles MMRd tumors were associated with a higher probability of metastasis than MMRp tumors [26,72,89]; 75% of patients with metastasis had MMRd tumors compared with 11.5% of patients with MMRp tumors (OR: 7.44).

### 3.3. IHC in the Theragnostics of MMRd Tumors in EC

We identified eight articles which investigated the role of IHC in the theragnostics for MMRd tumors in EC [53,88,93,95,122,136,161,162].

The studies included an average of 60.3 samples of MMRd tumors (minimum two, maximum 203).

PD-L1 expression is presented in Table 6.

Immune checkpoint pathways such as PD-1/PD-L1 are the main target for immunotherapy in EC. We identified four articles studying PD-L1 expression in MMRd tumors [88,95,136,161]. All the articles reported an increase in the expression of PD-L1 ranging from 60.4% to 100% in the peritumoral or tumoral compartment in MMRd tumors. Only 5.3% of MMRp tumors were reported to have PD-L1 expression.

One article reported a study of adjuvant radiotherapy in non-endometrioid MMRd tumors and showed a progression-free survival (PFS) and OS of 5 years compared with a PFS and OS of 2 years for non-endometrioid MMRp tumors [53].

We identified one article showing that MMRd tumors were associated with a significantly lower mean percentage of androgen receptor (AR) expression [122]; AR expression was 65.9% in MMRd tumors compared with 81.6% in MMRp tumors.

Lastly, one article showed that MMRp tumors were more sensitive to progestin treatment; no patients with MMRd tumors showed disease regression compared with 41% of patient with MMRp tumors [93].

## 4. Discussion

This is the first systematic review of the value of IHC markers in EC with MMRd tumors. IHC with expression of all four proteins and MLH1 promoter methylation remains the reference of choice for diagnosis because it is reproducible and applicable in routine clinical practice. The search for MMRd tumors is becoming essential for the care management of patients with EC regarding the diagnostic, prognostic, and theragnostic evaluations. We identified 10 articles using IHC expression of two proteins of the MMR repair system in EC with a detection rate from 13.5% to 100%, 18 articles using IHC expression of three proteins with a detection rate from 20% to 100%, and 96 articles using IHC expression of four proteins with a detection rate from 5% to 100%. Fifty-seven articles analyzed MLH1 promoter methylation with a detection rate from 14% to 100%. Overall, most of the articles suggest a better prognosis for MMRd tumors versus MMRp tumors. After a median follow-up of 31 months (1–99 months), there was no difference in progression or recurrence rates between pMMR and dMMR tumors (19.5% vs. 16.5%; *p* = 0.31). However, among those with non-endometrioid tumors, recurrence and mortality rates were significantly higher for pMMR than dMMR tumors (42.0% vs. 10.0%, *p* = 0.001, and 36.1% vs. 13.1%, *p* = 0.01, respectively), despite similar stage and lymphovascular space invasion distributions. Lastly, in the four articles studying PD-L1, all reported increased expression in MMRd tumors ranging from 60.4% to 100%.

In the early 1990s, and in the context of hereditary nonpolyposis colorectal cancer (HNPCC) development, the main focus was on mutations in MLH1 and MSH2 genes [163]. It was first thought that the four MMR proteins functioned as a heterodimer complex only; in the absence of one of the two proteins (MLH1 with PMS2 and MSH2 with MSH6), the complex no longer functions and the other protein is not expressed [164,165,166]. This explains why oncology departments initially routinely used two proteins alone in IHC analysis and not four. In our review, articles analyzing the expression of two proteins included an average of 17.9 MMRd tumors, whereas articles analyzing the expression of three and four proteins included an average of 34.6 and 98.5 MMRd tumors, respectively. Thus, the use of four proteins in IHC increases the number of MMRd cases identified. In addition, Goodfellow et al., in a cohort of 1002 patients with EC, showed that the most common MMR defect was MLH1 loss followed by combined MSH2/MSH6 losses, then MSH6 loss alone at 70%, 20.5%, and 19.6%, respectively [71]. Several articles highlighted an increased prevalence of MSH6 mutations representing 3.8% (95% CI 1.0–9.5%) of patients with EC compared to 2.6% (95% CI 0.5–7.4%) of patients with HNPCC tumors [167,168,169]. It is, thus, essential to screen patients with EC with IHC using the four proteins. Screening with only two proteins would not only underestimate the prevalence of MMRd tumors in a cohort but would also misclassify patients as having MMRp tumors which would result in mistreatment.

As mentioned above, since 2018, the National Comprehensive Cancer Network recommends universal MMR testing for all newly diagnosed cases of EC [170]. Indeed, Mills et al. reported that 57.1% of patients with LS would not be identified on the basis of age and individual cancer history alone, and that 28.6% would not be identified even with a complete family history [82], implying that the Bethesda criteria are insufficient [171,172]. IHC has since become the standard practice in many institutions and is recommended in new guidelines as the gold standard screening test. As a consequence, recent articles studying the cost effectiveness of reflex testing for LS-related EC in patients older than 70 years that do not meet the Bethesda criteria suggest that the most cost-effective approach is to test all EC patients up to an age threshold (somewhere between 60 and 65 years) and that even testing all patients up to 70 or 80 years would be cost-effective compared with no testing. Using a cost-effectiveness threshold of 20,000 GBP per quality-adjusted-life-year (QALY), reflex testing for LS using MMR IHC and MLH1 methylation testing was cost-effective versus no testing, costing 14,200 GBP per QALY gained [173].

However, several other techniques, including molecular biology, are increasingly being studied. In our systematic review, we identified 96 studies including both IHC and MSI molecular analysis. Among these, 87 used a PCR-based molecular testing using five of the eight mononucleotide or dinucleotide repeats (BAT-25, BAT-26, NR-21, NR-24, NR-27, D5S346, D2S123, and D17S250), 15 used NGS, and two used the Idylla MSI test [174,175]. Sixteen articles reported MSI-High tumors when more than two markers were unstable, and two articles reported MSI-High when more than three markers were unstable. However, researchers have found other techniques to overcome the drawbacks of IHC. Stello et al., in large, randomized cohort trials in EC such as PORTEC-1 and PORTEC-2, concluded that MSI and IHC analyses are highly concordant (94%) [75]. In the same way, Stinton et al. showed no statistically significant differences in test accuracy estimates (sensitivity and specificity); the sensitivity of IHC ranged from 60.7–100%, and the sensitivity of MSI-based testing ranged from 41.7–100% [137]. Therefore, IHC remains the cheapest and most accurate test with a low failure rate to determine MMR. However, it would be interesting to find a new technique for the detection of MMRd tumors using artificial intelligence and molecular biology that is more manageable, faster, more reproducible, and less expensive than IHC.

IHC is essential to determine the prognosis for patients with certain tumors. In 2021, the ESMO described prognostic risks groups according to the stage, grade, lymphovascular involvement, myometrial invasion, and the molecular classification obtained by IHC to establish clinical and therapeutic guidelines [12].

In our systematic review, we identified 34 articles studying the prognosis in MMRd/MMRp tumors. A larger number of articles demonstrated a better RFS and a better OS for patients with MMRd tumors versus MMRp tumors, but these articles were based on a small number of patients. The article by Pina et al., consisting of 242 MMRd tumors and representing the largest MMRd cohort studying prognosis in our systematic review, showed that, among non-endometrioid tumors, recurrence and mortality rates were significantly higher for pMMR than dMMR tumors (42.0% vs. 10.0%, *p* = 0.001, and 36.1% vs. 13.1%, *p* = 0.01, respectively), despite similar stage and lymphovascular space invasion distributions [99]. We also identified controversial findings regarding the age at diagnosis, the stage, and the lymph node status of MMRd tumors. Studying 212 MMRd tumors, Tangjitgamol et al. found a significantly higher rate of MMRd tumors in patients aged less than 60 years, with early-stage disease, and more negative lymph node status than the other comparative groups: 59.2% vs. 48.3% (*p* = 0.037) for age, 58.2% vs. 45.2% (*p* = 0.027) for stage, and 58.1% vs. 44.6% (*p* = 0.048) for nodal status [92]. Despite studies to the contrary, new studies tend to describe MMRd tumors as tumors with a good prognosis compared to MMRp tumors. Knowledge of the MMR status is, therefore, essential for the therapeutic management of patients with EC.

Theragnostic studies are currently fashionable, and theragnostics are being increasingly used in patient management. In our systematic review, we identified eight articles studying the theragnostics of MMRd tumors in EC. All showed an increased expression, of up to 100%, of PD-L1 tumoral compartment in MMRd tumors compared to 5.3% in MMRp tumors. Currently, it is well established that PD-1 and PD-L1 checkpoints are the main target in immunotherapy for recurrence or refractory cases in several cancers [176,177]. Compared to other gynecological cancers, EC shows a large number of immune cells and intra-tumoral cytokines, which stimulate an endogenous antitumor immune response [178,179]. In a phase Ib/II clinical trial including patients with advanced EC receiving lenvatinib (a tyrosine kinase inhibitor) plus pembrolizumab, the objective response rate was 63.9% (95% CI: 30.8–80.9%) for MMRd tumors compared with 37.2% (95% CI: 37.5–47.8%) for MMRp tumors [180]. Immunotherapy for patients with EC, especially in cases of advanced or metastatic disease with MMRd tumors, is attracting considerable attention as more than 50 clinical trials investigating immunotherapy in EC have been listed on the clinicaltrials.gov website [181]. To date, three monoclonal antibodies targeting PD-1 (pembrolizumab, nivolumab, and cemiplimab) and three monoclonal antibodies targeting PD-L1 (atezolizumab, durvalumab, and avelumab) have been approved by the US Food and Drug Administration for advanced inoperable cancers in first-line, metastatic, and recurrent EC.

Our systematic review confirms Jumaah et al.’s meta-analysis which showed a large variation in the diagnosis of MMRds tumors. This depends on the initial study population, which may include only low- or high-grade tumors, Lynch families, or concurrent cancers. In contrast to us, this meta-analysis included MMRd tumors diagnosed by molecular biology, which probably allowed them to obtain a larger variation. From a prognostic point of view, our articles are concordant since they point out that MMRd tumors have a more favorable prognosis compared with MMR-proficient tumors, and that their immune context is a key point of immunotherapy [182].

This systematic review had some specific limitations. First, many studies did not detail their method of analysis for IHC, and the cohorts were heterogeneous, which made it impossible to synthesize the results in the form of a meta-analysis. The second limitation was the difficulty of comparing diagnostic techniques in terms of therapeutic effectiveness and cost-effectiveness. The strengths of our study were that the methodology used with the PRISMA Guidelines, and we included more than 161 articles classified by diagnosis, prognosis, and theragnostics. This review of the literature allows us to state that, in order not to underestimate the MSI tumors in the EC, it is strongly recommended to use the four proteins of the MMR system through the IHC technique, which remains the least expensive and most reproducible. Furthermore, our review updates the data in the literature on the prognosis of these tumors. The research on MSI tumors tends toward a better prognosis thanks to their particular immune context, which is key to many immunotherapies.

## 5. Conclusions

In this systematic review, we provided an overview of MMR status through IHC in EC. IHC with expression of all four proteins and MLH1 promoter methylation remains the reference of choice for diagnosis because it is reproducible, cheaper, and applicable in routine clinical practice. Molecular classification such as MMRd tumors has been essential to determine the prognosis. More and more clinical trials are using the immune context of MMRd tumors in immunotherapy treatment. Further studies are needed to evaluate IHC in comparison with molecular tests including artificial intelligence, in terms of both efficacy and medical/economic aspects.

## Figures and Tables

**Figure 1 cancers-14-03783-f001:**
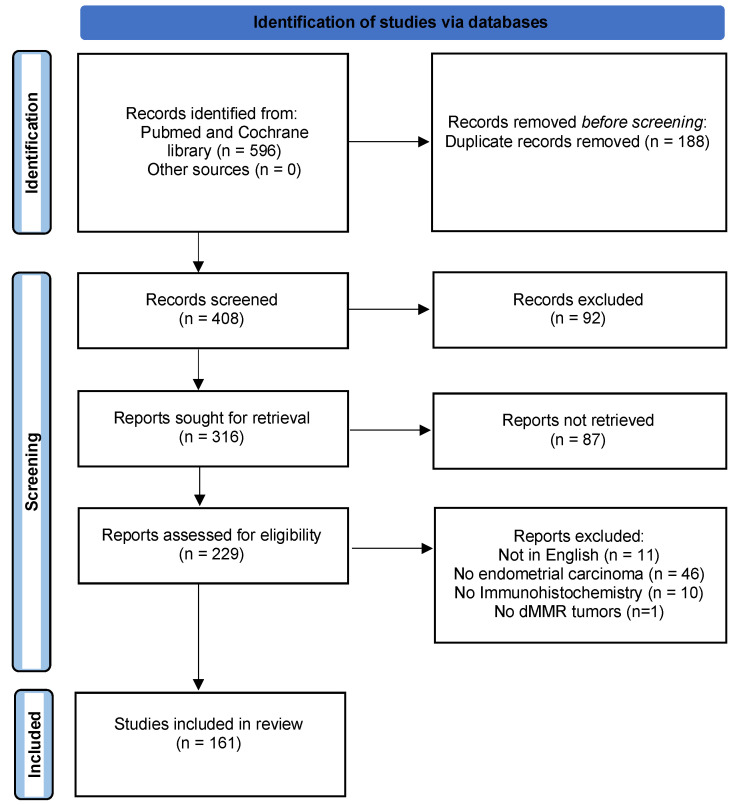
PRISMA flowchart.

**Table 1 cancers-14-03783-t001:** Differences in immunohistochemistry protein losses in tumors with microinstability in endometrial cancer using the expression of two proteins.

Reference	Type of Tissue	Germinal Mutation Included	MMRd (*n*)	MMRd (%)	MLH1 (%)	MSH2 (%)	MSH6 (%)	PMS2 (%)	MLH1 Promoter Methylation (%)	
**Simpkins** 1999 [19]	FPPE + frozen tissue	0	53	100	14.3	86	-	-	77	
**Peiro** 2001 [20]	FFPE	0	12	13.5	12.4	1.1	-	-	-	
**Maruyama** 2001 [21]	FFPE	1	31	37	12	19	-	-	-	
**Berends** 2001 [22]	FFPE	1	13	27	61.5	30.7	-	-	-	
**Chiaravalli** 2001 [23]	FFPE	1	13	39.4	-	-	-	-	-	
**Hardisson** 2003 [24]	FFPE	0	23	29.5	73.9	21.7	-	-	-	
**Sutter** 2004 [25]	Frozen tissue	1	13	39.4	20	76.9	-	-	-	
**Irving** 2005 [26]	FFPE	0	6	50	83.3	16.7	-	-	83	
**Alvarez** 2012 [27]	FFPE + frozen Tissue	0	4	16	75	25	-	-	-	
**Plotkin** 2020 [28]	FFPE	0	11	22	-	-	0	100	-	

FFPE: formalin-fixed paraffin-embedded, MMRd: mismatch repair-deficient.

**Table 2 cancers-14-03783-t002:** Differences in immunohistochemistry protein losses in tumors with microinstability in endometrial cancer using the expression of three proteins.

Reference	Type of Tissue	Germinal Mutation Included	MMRd (*n*)	MMRd (%)	MLH1 (%)	MSH2 (%)	MSH6 (%)	PMS2 (%)	MLH1 Promoter Methylation
**Planck** 2002 [29]	FFPE	1	21	-	9.5	47.6	57.1	-	-
**Orbo** 2003 [30]	FFPE	0	18	-	55.5	22.2	33.3	-	-
**Lipton** 2004 [31]	Healthy tissue	1	67	27	44.8	43.3	1.5	-	-
**Macdonald** 2004 [32]	FFPE + frozen	0	164	51.9	14	19	17	-	69
**Buttin** 2004 [33]	Frozen tissue + healthy tissue	1	94	22.8	-	-	-	-	70
**Soliman** 2005 [34]	FPPE	1	12	20	58.3	41.7	41.7	-	-
**Cederquist** 2005 [35]	FFPE	1	6	100	0	0	66.7	-	-
**Ollikainen** 2005 [36]	FFPE + healthy tissue	1	16	48.5	43.8	25	31.3	-	-
**Taylor** 2006 [37]	FPPE	0	6	21	33.3	0	0	-	-
**Niessen** 2006 [38]	Not described	0	36	17.1	30.6	13.9	55.6	-	-
**Rijcken** 2006 [39]	FFPE	1	18	100	33.3	11.1	11.1	-	-
**Yoon** 2008 [40]	FFPE + healthy tissue	1	50	44.2	46	4	2	-	14
**Arabi** 2009 [41]	FFPE + healthy tissue	0	25	21	22.5	18	43	-	-
**Walsh** 2010 [42]	FFPE	1	9	12.5	12	2	6	-	-
**Yasue** 2011 [43]	FFPE + frozen tissue	0	8	22.9	100	25	45.8	-	-
**Huang** 2014 [44]	FFPE	1	29	-	27.6	55.2	17.2	-	-
**Kobayashi** 2015 [45]	FFPE	0	17	53.1	82.4	11.8	64.7	-	-
**Ren** 2020 [46]	FFPE	1	27	12.8	38.1	37	51.8	-	33.3

FFPE: formalin-fixed paraffin-embedded, MMRd: mismatch repair-deficient.

**Table 3 cancers-14-03783-t003:** Differences in immunohistochemistry protein losses in tumors with microinstability in endometrial cancer using expression of four proteins.

Reference	Type of Tissue	Germinal Mutation Included	MMRd (*n*)	MMRd (%)	MLH1 (%)	MSH2 (%)	MSH6 (%)	PMS2 (%)	MLH1 Promoter Methylation (%)
**Westin** 2008 [47]	FFPE	1	12	34.3	25	75	75	25	16.7
**Matthews** 2008 [48]	FFPE + healthy tissue	1	21	34.4	85.7	95.3	23.8	76.2	-
**Garg** 2009 [49]	FFPE	0	9	20	33.3	33.3	50	44.4	-
**Garg** 2009 [50]	FFPE	1	32	45	59.4	40.6	40.6	59.4	-
**Tafe** 2010 [51]	Not described	1	8	47	87.5	0	12.5	87.5	-
**Cossio** 2010 [52]	FFPE + healthy tissue	1	7	30	28.6	14.3	71.4	28.6	-
**Resnick** 2010 [53]	FFPE	0	155	66.5		-	-	-	-
**Shih** 2011 [54]	FFPE	1	9	16.1	44.4	55.6	55.6	44.4	-
**Leenen** 2012 [55]	FFPE	1	42	23.5	76.2	23.8	23.8	23.8	73.8
**Soslow** 2012 [56]	FFPE	0	7	31.8	100	0	0	100	-
**Egoavil** 2013 [57]	FFPE	1	61	33.5	72.1	8.2	21.3	13.1	55.7
**Bosse** 2013 [58]	FFPE	1	36	24.7	-	-	-	-	88.9
**Moline** 2013 [59]	FFPE	1	59	24.1	84.7	15.3	18.6	15.3	55.9
**Peiro** 2013 [60]	FFPE	1	63	24.4	79.4	4.8	17.6	79.4	-
**Romero-Perez** 2013 [61]	FFPE	0	39	32.5	48.7	23.1	30.8	69.2	-
**Mills** 2014 [62]	FFPE	1	137	22.6	72.3	27	27	72.3	-
**Ruiz** 2014 [63]	FFPE	0	64	30.2	54.7	6.3	54.7	56.3	-
**Thoury** 2014 [64]	FFPE + healthy tissue	0	17	24.6	65	0	23	59	-
**Rabban** 2014 [65]	FFPE	1	41	15	75.6	7.7	17	7.3	-
**Long** 2014 [66]	Not described	1	41	23.7	24.4	51.2	68.3	31.7	-
**Woo** 2014 [67]	FFPE	0	15	19.5	86.7	13.3	13.3	86.7	-
**Hoang** 2014 [68]	FFPE	0	6	9.5	50	16.7	16.7	83.3	-
**Buchanan** 2014 [69]	FFPE	1	170	24	75	13	24.7	75.3	-
**Allo** 2014 [70]	FFPE	0	63	33	73	15.9	23.8	73	-
**Goodfellow** 2015 [71]	FFPE + Frozen tissue	1	360	38.4	75.2	3.1	11.9	7.5	70.3
**Chu** 2015 [72]	FFPE + frozen tissue + healthy tissue	0	22	32.8	27.2	22.7	72.3	27.2	-
**Graham** 2015 [73]	FFPE + healthy tissue	1	-	-	-	-	-	-	-
**Dudley** 2015 [74]	FFPE	1	72	33.4	-	-	-	20.8	-
**Stelloo** 2015 [75]	FFPE	0	19	16.4	63.2	-	-	-	47.4
**Mao** 2015 [76]	Not described	0	19	46.3	73.7	10.5	15.8	78.9	-
**Mc Conechy** 2015 [77]	Frozen tissue	0	38	24.2	50	5.3	10.5	55.3	-
**Stewart** 2015 [78]	FFPE	0	13	59.1	76.9	15.4	30.8	84.7	-
**Watkins** 2016 [79]	FFPE	1	27	21.6	88.9	3.7	14.8	44.4	25.9
**Pocrnich** 2016 [80]	Not described	1	8	44.4	75	12.5	12.5	75	-
**Lin** 2016 [81]	Not described	1	17	22.3	82.3	17.6	17.6	82.3	64.7
**Mills** 2016 [82]	Not described	1	66	31.4	65.2	18.2	31.8	71.2	-
**Ramalingam** 2016 [83]	FFPE	0	18	51.4	94.4	5.6	5.6	94.4	-
**Shikama** 2016 [84]	FFPE	1	62	28	62.9	14.5	38.7	67.7	-
**Kato** 2016 [85]	FFPE	1	8	2.2	-	-	-	-	-
**Okoye** 2016 [86]	FFPE	0	40	9.7	-	-	-	-	75
**Russo** 2017 [87]	FFPE + healthy tissue	0	3	50	66.7	33.3	33.3	66.7	-
**Bregar** 2017 [88]	FFPE	0	13	18.5	-	0	0	-	-
**Pelletier** 2017 [89]	Not described	0	34	26.8	67.6	8.8	14.7	85.3	-
**Stelloo** 2017 [90]	FFPE	0	169	24.3	85.2	5.9	13.6	89.9	-
**Dillon** 2017 [91]	FFPE	1	60	26	85	3.3	3.3	85	81.2
**Tangjitgamol** 2017 [92]	FFPE	0	212	55.1	60.4	29.7	70.3	62.3	-
**Zakhour** 2017 [93]	FFPE	1	6	7.1	33.3	50	66.7	33.3	-
**Najdawi** 2017 [94]	FFPE	1	36	29	86.7	13.9	26.7	93.3	-
**Sloan** 2017 [95]	FFPE	1	38	56.7	-	-	-	-	15.8
**Watkins** 2017 [96]	FFPE	1	48	19.8	81.3	8.3	14.6	85.4	-
**Chen** 2017 [97]	FFPE	0	30	10.3	-	-	-	-	-
**Kobel** 2017 [98]	FFPE	0	6	37.5	50	33.3	50	33.3	-
**Pina** 2018 [99]	Not described	1	242	27.1	78.9	21.1	21.1	78.9	69
**Adar** 2018 [100]	Not described	1	107	22.1	80.3	19.6	19.6	15.4	70
**Chapel** 2018 [101]	FFPE	1	30	30.3	86.7	10	10	90	-
**Bosse** 2018 [102]	Not described	0	136	36.2	-	-	-	-	-
**Saita** 2018 [103]	Not described	1	13	-	46.2	23.1	30.8	-	-
**Espinosa** 2018 [104]	FFPE	0	2	50	100	0	0	100	-
**Li** 2018 [105]	FFPE	0	162	23.1	80.2	19.8	19.8	80.2	-
**Doghri** 2019 [106]	FFPE	0	10	22.2	80	10	10	80	-
**Hashmi** 2019 [107]	FFPE	1	56	44.4	92.9	17.9	35.7	89.3	-
**Saeki** 2019 [108]	FFPE + healthy tissue	0	18	18.4	77.8	22.2	44.4	83.3	-
**Zannoni** 2019 [109]	FFPE	0	15	33.3	0	46.7	73.3	26.7	-
**Abdufatah** 2019 [110]	Not described	0	20	40	5	10	80	80	-
**Chapel** 2019 [111]	FFPE	1	17	100	94.1	5.9	5.9	94.1	-
**Kahn** 2019 [112]	Not described	1	1672	28.3	69.3	-	-	-	53.9
**Ryan** 2019 [113]	Not described	1	2563	24.5	17.5	3	3	2	37.6
**Wu** 2019 [114]	FFPE	1	50	100	63	10	24	72	-
**Saijo** 2019 [115]	FFPE	0	6	10.5	-	-	-	-	-
**Sarode** 2019 [116]	FFPE	1	45	9.3	-	-	-	-	26.7
**Sari** 2019 [117]	FFPE	0	22	30	68.2	9	13.6	68.2	-
**Lucas** 2019 [118]	FFPE	1	63	-	54	19	42.9	55.6	-
**Baniak** 2019 [119]	FFPE	0	0	-	-	-	-	-	-
**Backes** 2019 [120]	Not described	0	64	32.5	-	-	-	-	-
**Dong** 2019 [121]	FFPE	0	63	24	-	-	-	-	-
**Gan** 2019 [122]	FFPE	1	91	27.2	87.9	12.1	12.1	87.9	69.2
**He** 2019 [123]	Not described	1	2	3.3					-
**Ryan** 2020 [124]	FFPE + healthy tissue	1	132	26	75.8	9.1	18.9	75.8	62.9
**Rosa** 2020 [125]	FFPE	1	80	33.1	51.3	12.5	22.5	43.8	48.8
**Beinse** 2020 [126]	FFPE	0	35	29.7	-	-	17	83	-
**Timmerman** 2020 [127]	Not described	1	33	31	81.8	3	15.2	6.1	79
**Missaoui** 2020 [128]	FFPE	1	1	3.7	100	-	-	100	100
**Dasgupta** 2020 [129]	Not described	1	4	-	100	-	-	100	-
**Kolehmainen** 2020 [130]	FFPE	0	287	47.5	-	-	-	-	-
**León-Castillo** 2020 [131]	FFPE	0	137	33.4	-	-	-	-	-
**Rekhi** 2020 [132]	FFPE	0	50	-	66	28	28	66	-
**Kim** 2020 [133]	FFPE	0	5	9.6	-	-	-	-	-
**Pasanen** 2020 [134]	FFPE + tumor cells	0	191	37.3	-	-	-	-	-
**Jin** 2020 [135]	Not described	0	1	5	-	-	-	-	-
**Rowe** 2020 [136]	FFPE	1	43	-	-	-	-	-	46.5
**Stinton** 2021 [137]	Not described	1	-	-	-	-	-	-	-
**Pecriaux** 2021 [138]	FFPE	1	9	60	100	0	0	88.9	-
**Tjalsma** 2021 [139]	Not described	1	41	23	14	9	9	14	14
**Joehlin-Price** 2021 [140]	FFPE	1	35	36.8	77.1	11.4	25.9	77.1	20
**Yamamoto** 2021 [141]	FFPE	1	68	17.2	77.9	16.2	17.6	79.4	75

FFPE: formalin-fixed paraffin-embedded, MMRd: mismatch repair-deficient.

**Table 4 cancers-14-03783-t004:** Differences in MLH1 promoter methylation.

Reference	Technique	MLH1 Promoter Methylation (%)
**Simpkins** 1999 [19]	PCR + bisulfite conversion	14.3
**Horowitz** 2002 [143]	PCR + bisulfite conversion	-
**Strazzullo** 2003 [144]	PCR	-
**Buttin** 2004 [33]	PCR + bisulfite conversion	70
**Macdonald** 2004 [32]	Not described	69
**Soliman** 2005 [34]	PCR + bisulfite conversion	-
**Irving** 2005 [26]	PCR	83.3
**Kanaya** 2005 [145]	PCR + bisulfite conversion	-
**Ollikainen** 2005 [36]	PCR	-
**Westin** 2008 [47]	PCR	16.7
**Nam Yoon** 2008 [40]	PCR	14
**Zighelboim** 2009 [146]	Pyrosequencing and/or combined bisulfite restriction analysis	
**Koyamatsu** 2010 [147]	Not described	-
**Walsh** 2010 [42]	PCR + bisulfite conversion	-
**Cossio** 2010 [52]	Methylation-specific multiplex ligation-dependent probe amplification	
**Leenen** 2012 [55]	Methylation-specific multiplex ligation-dependent probe amplification	73.8
**Egoavil** 2013 [57]	Methylation-specific multiplex ligation-dependent probe amplification	55.7
**Bosse** 2013 [58]	PCR + bisulfite conversion	88.9
**Moline** 2013 [59]	PCR	55.9
**Batte** 2014 [148]	Not described	-
**Bruegl** 2014 [149]	PCR	-
**Buchanan** 2014 [69]	PCR + bisulfite conversion	
**Goodfellow** 2015 [71]	Pyrosequencing and/or combined bisulfite restriction analysis	70.3
**Stelloo** 2015 [75]	PCR	47.4
**McConechy** 2015 [77]	PCR	-
**Goverde** 2016 [150]	Not described	-
**Watkins** 2016 [79]	PCR	25.9
**Lin** 2016 [81]	PCR	64.7
**Mills** 2016 [82]	PCR	-
**Ramalingam** 2016 [83]	PCR	-
**Shikama** 2016 [84]	PCR + bisulfite conversion	-
**Kato** 2016 [85]	Methylation-specific multiplex ligation-dependent probe amplification	-
**Okoye** 2016 [86]	PCR + bisulfite conversion	75
**Bruegl** 2017 [151]	PCR	-
**Zeimet** 2017 [152]	Not described	-
**Stelloo** 2017 [90]	PCR	-
**Dillon** 2017 [91]	PCR	81.2
**Najdawi** 2017 [94]	PCR	-
**Sloan** 2017 [95]	PCR	15.8
**Watkins** 2017 [96]	PCR + bisulfite conversion	-
**Adar** 2018 [100]	PCR + bisulfite conversion	70
**Pina** 2018 [99]	Not described	69
**Kahn** 2019 [112]	Not described	53.9
**Ryan** 2019 [113]	Not described	37.6
**Sarode** 2019 [116]	Not described	26.7
**Gan** 2019 [122]	PCR	69.2
**Ryan** 2020 [124]	NGS for germline mutation	62.9
**Rosa** 2020 [125]	PCR + bisulfite conversion + NGS for germline mutation	48.8
**Timmerman** 2020 [127]	NGS for germline mutation	79
**Missaoui** 2020 [128]	PCR + bisulfite conversion	100
**Dasgupta** 2020 [129]	Methylation-specific multiplex ligation-dependent probe amplification	-
**Ren** 2020 [46]	PCR + bisulfite conversion	33.3
**Rowe** 2020 [136]	PCR	46.5
**Stinton** 2021 [137]	Not described	-
**Tjalsma** 2021 [139]	Not described	14
**Joehlin-Price** 2021 [140]	Not described	20
**Yamamoto** 2021 [141]	PCR	75

PCR: polymerase chain reaction.

**Table 5 cancers-14-03783-t005:** Prognosis of MMRd tumors compared with MMRp tumors in EC.

Reference	EC Total (*n*)	Type of Tissue	MMRd (*n*)	* RFS or Recurrence MMRd	* RFS or Recurrence MMRp	** OS or Deaths MMRd	** OS or DeathsMMRp	Prognosis Conclusion
**Parc** 2000 [153]	62	Fresh + normal tissue	21	-	-	-	-	Not significant
**Ju** 2006 [154]	50	FFPE	12	-	-	-	-	Not significant
**Choi** 2008 [155]	39	FFPE	8	-	-	-	-	Not significant
**Shih** 2011 [54]	56	FFPE	9	5 year: 71.1%, 95% CI (53.1–89.1%)	5 year: 97.6%, 95% CI (95.2–100%)	5 year: 71.1%, 95% CI (53.1–89.1%)	5 year: 100%	MMRd was associated with worse RFS and OS compared with MMRp
**Peiro** 2013 [20]	260	FFPE	33	-	-	-	-	Not significant
**Ruiz** 2014 [63]	212	FFPE	64	-	-	-	-	Not significant
**Stelloo** 2015 [75]	216	FFPE	19	5 year: 95%	5 year: 93% POL-E, 52% no specific molecular profile 42% p53	-	-	MMRd was associated with a better RFS compared with MMRp
**Mao** 2015 [76]	41	Not described	16	-	-	-	-	Not significant
**Tangjitgamol** 2017 [92]	385	FFPE	212	5 year: 67.0%, 95% CI (49.7–79.5%) advanced stage	5 year: 40.0%, 95% CI (25.5–54.1%) advanced stage	5 year: 66.5%, 95% CI (49.2–79.1) advanced stage	5 year: 45.5%, 95% CI (30.2–59.5) advanced stage	MMRd was associated with better RFS and OS in advanced stage compared with MMRp in advanced stage
**Pina** 2018 [99]	892	Not described	242	Recurrence: 10%	Recurrence: 42%	Deaths: 13.1%	Deaths: 36.1%	MMRd was associated with better RFS and OS compared with MMRp
**Bosse** 2018 [102]	381	Not described	136	HR = 0.61, 95% CI (0.37–1.00) compared with no specific molecular profile	HR = 0.23, 95% CI (0.07–0.77) POLE compared with no specific molecular profile	HR = 0.84, 95% CI (0.57–1.25) compared with no specific molecular profile	HR = 0.56, 95% CI (0.27–1.15) POLE compared with no specific molecular profile	MMRd was associated with better RFS and OS compared with MMRp
**Backes** 2019 [120]	197	Not described	64	5 year: 66%, 95% CI (45–79%)	5 year: 89%, 95% CI (76–94%)	5 year: 74%	5 year: 86%	MMRd was associated with worse RFS and OS compared with MMRp
**Beinse** 2020 [126]	159	FFPE	35	-	-	-	-	Intermediate
**Kolin** 2020 [158]	96	FFPE	34	36 months	9 months	-	-	MMRd was associated with better RFS compared with MMRp
**León-Castillo** 2020 [131]	423	FFPE	137	5 year: 71.7%	5 year: 98% POL-E, 48% p53	5 year: 81.3%	5 year: 98% POLE, 53% p53	Intermediate
**Kim** 2020 [133]	52	FFPE	5	-	-	-	-	Not significant
**Joehlin-Price** 2021 [140]	95	FFPE	35	-	-	-	-	Not significant
**Yamamoto** 2021 [141]	395	FFPE	68	-	-	-	-	Not significant

FFPE: formalin-fixed paraffin-embedded, MMRd: mismatch repair-deficient, MMRp: mismatch repair-proficient, RFS: recurrence-free survival, OS: overall survival. * RFS or recurrence is presented according to the data available; ** OS or death is presented according to the data available, HR: hazard ratio; POLE: polymerase-ε, CI: confidence interval.

**Table 6 cancers-14-03783-t006:** PDL-1 in MMRd tumors in endometrial cancer.

Reference	Type of Tissue	MMRd (*n*)	MMRd (%)	PDL-1
**Jones** 2020 [161]	Not described	203	33	PDL-1 was more frequent in MMRd tumors
**Bregar** 2017 [88]	FFPE	13	33	PDL-1 was present in 62% of MMRd tumors and high-grade tumors vs. 46% in MMRp tumors
**Sloan** 2017 [95]	FFPE	38	56.7	100% MMRd tumors demonstrated PDL-1 expression in peritumoral immune compartment
**Rowe** 2020 [136]	FFPE	43	69.4	60.4% MMRd tumors showed positive tumoral PDL-1 vs. 5.3% MMRp

FFPE: formalin-fixed paraffin-embedded, MMRd: mismatch repair-deficient.

## Data Availability

No new data were created or analyzed in this study. Data sharing is not applicable to this article.

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
