# Peer review of "The Role of Immunohistochemistry Markers in Endometrial Cancer with Mismatch Repair Deficiency: A Systematic Review"

_cancers, 2022, doi:10.3390/cancers14153783_

Round 1

Reviewer 1 Report

The manuscript was prepared very well. The introduction section justifies the purpose of the study. I congratulate the authors for the preparation of the manuscript

However, I have the following comments:

Introduction

·       There are several sentences and paragraphs without reference, add them.

·       Discuss the importance and differences of IHC markers and other markers used in endometrial cancer

Methods

·       I recommend that you register the review in PROSPERO.

·       Indicate in an Annex the search sequence used, adding the Boolean operators used.

·       Indicate in an Annex the search sequence used, adding the Boolean operators used.

·       I recommend using an evaluation of the methodological quality of the articles chosen: McMaster, PEDRo...

Results

Discussion

·       What is new in this manuscript for IHC cancer?

·       Include a section on strengths.

·       What does this article contribute to, the authors should make their own assessment and include their own discussion of the results shown in the manuscript?

·       In the Conclusion section, state the most important outcome of your work. Do not simply summarize the points already made in the body — instead, interpret your findings at a higher level of abstraction. Show whether, or to what extent, you have succeeded in addressing the need stated in the Introduction (or objectives).

Author Response

Reviewer 1

1

The manuscript was prepared very well. The introduction section justifies the purpose of the study. I congratulate the authors for the preparation of the manuscript

Thank you for this comment.

2

There are several sentences and paragraphs without reference, add them.

We agree with this comment.

We added reference in the introduction.

“However, once the cancer spreads to distant organs, the prognosis is poor with a 5-year survival rate of 17% [3].”

“Furthermore, this specific molecular genetic alteration, resulting from a defect in the MMR genes hMLH1, hMSH2, hMSH6, or hPMS2 is found in sporadic tumors called MMRd or Lynch-like syndrome tumors [10].”

3

Discuss the importance and differences of IHC markers and other markers used in endometrial cancer

Thank you for this comment.

We added this paragraph In the introduction:

In addition to MMR testing and to better identify risk groups currently included in the latest European Society of Gynecological Oncology/European Society for Radiotherapy and Oncology/European Society of Pathology (ESGO/ESTRO/ESP) [12], IHC markers such as p53 or POL-E have been proposed in a diagnostic algorithm [13]. Recent studies showed that abnormal p53 IHC reliably identifies cases with TP53 mutation (representing 25% of all EC) in EC biopsies (94% specificity and 91% sensitivity) [14]. POL-E mutation reprensenting 8.59% of all EC are mainly presented at earlier stages I-II (89.51%) and at the highest grade III (51.53%) [15]. Other IHC markers have been studied such as oestrogen receptor, progesterone receptor, HER-2, ki67, however, no single marker was found to be indicative of EC often enough to allow routine use in the sub-classification of EC [16].

4

 I recommend that you register the review in PROSPERO.

Thank you for this comment.

We will add in prospero if the editor wishes.

5

Indicate in an Annex the search sequence used, adding the Boolean operators used.

We used the PRISMA guidelines.

there is the methodology used.

Nous l’avons mis dans la méthodologie et nous pourons le rajouter si l’éditeur le souhaite.

6

 I recommend using an evaluation of the methodological quality of the articles chosen: McMaster, PEDRo...

We used the PRISMA guidelines as recommended by the editor

We used the PRIMA guidelines as suggested by the editor. There is a checklist in the appendix, if you or the editor wish we can add it in the main text.

7

What is new in this manuscript for IHC cancer?

To the best of our knowledge, there is no recent review of the value of IHC markers to identify MMRd EC phenotypes or LS related EC.

This is the first study to focus primarily on iHC by evaluating the digansotic prognosis and theranostic.

8

Include a section on strengths.

Thank you for this comment

We added a paragraph in the discussion:

“The strengths of our study are the methodology used with the PRISMA Guidelines. In addition, we included 161 articles classified by diagnosis, prognosis and theragnostic.”

9

What does this article contribute to, the authors should make their own assessment and include their own discussion of the results shown in the manuscript?

Thank you for this comment.

We added a paragraph in the discussion:

“This is the first systematic review of the value of IHC markers in EC with MMRd tumors. IHC with expression of all four proteins and MLH1 promoter methylation remains the reference of choice for diagnosis because it is reproducible, applicable in routine clinical practice. The search for MMRd tumors is becoming essential for the care management of patients with EC regarding the diagnosis, prognostic and theragnostic. “

10

  In the Conclusion section, state the most important outcome of your work. Do not simply summarize the points already made in the body — instead, interpret your findings at a higher level of abstraction. Show whether, or to what extent, you have succeeded in addressing the need stated in the Introduction (or objectives).

Thank you for this comment.

We change our conclusion:

“In this systematic review, we provide an overview of MMR status through IHC in EC. IHC with expression of all four proteins and MLH1 promoter methylation remains the reference of choice for diagnosis because it is reproducible, cheaper and applicable in routine clinical practice. Molecular classification such as MMRd tumors has been essential to determine the prognosis. More and more clinical trials are using the immune context of MMRds tumors in immunotherapy treatment.”

Reviewer 2 Report

In the present systematic review titled “The role of immunohistochemistry markers in endometrial cancer with mismatch repair deficiency:  a systematic review”, the authors have evaluated the clinicopathologic features of patients with mismatch repair deficiecy (MMRd) in order to better understand this increasing population.

1.     The review should be more precise in the inclusion criteria and state if there were any specific exclusion criteria. The materials and methods section should be written more robustly.

2.     The analysis is completely focused only on the study of MMR-deficient patients who are part of a 15-30% slice of all endometrial cancers, leaving out ECs with MMR-intact, p53-mutant and POLE-mutant. This intent is stated by the authors, so it is one of the main inclusion criteria. However, these molecular subgroups are also reported in the results. Authors should better specify whether patients with Lynch syndrome are included in the selected papers.

3.     The authors should report a table with overall patient demographics and clinicopathologic findings (age range, percentage of stage and grade and histology of tumors).

4.     The authors did not carry out association analyses, however, apart from the non-homogeneity of the samples, some associative aspects between the clinical characteristics and the markers taken into consideration should be analyzed through different statistical tests. At least select the studies with which the authors believe they can do a meta-analysis.

The risk is that this manuscript will remain a mere sequence of lists of papers with qualitative conclusions. The scientific level that the journal has reached requires this effort.

5.     In this recent paper: Jumaah AS, Al-Haddad HS, Salem MM, McAllister KA, Yasseen AA. Mismatch repair deficiency and clinicopathological characteristics in endometrial carcinoma: a systematic review and meta-analysis. J Pathol Transl Med. 2021 May;55(3):202-211. doi: 10.4132/jptm.2021.02.19, the authors conclude that the frequency of MMR deficiency in EC remains elusive. The authors should provide comment on any inconsistencies with their results.

Author Response

Reviewer 2

1

The review should be more precise in the inclusion criteria and state if there were any specific exclusion criteria. The materials and methods section should be written more robustly.

Thank you for this comment.

We added the exclusion criteria:

The exclusion criteria were: no detail of IHC, no MMRd tumors, no detail of EC, MMRd diagnose with molecular approach only.”

Considering the robustness of the methodology in agreement with the editor we have made the prisma checklist which can be found in the annex. If you or the editor wish we can transfer a part of this appendix into the methodology.

2

The analysis is completely focused only on the study of MMR-deficient patients who are part of a 15-30% slice of all endometrial cancers, leaving out ECs with MMR-intact, p53-mutant and POLE-mutant. This intent is stated by the authors, so it is one of the main inclusion criteria. However, these molecular subgroups are also reported in the results. Authors should better specify whether patients with Lynch syndrome are included in the selected papers.

Thank you for this comment.

We added: “Lynch syndrome patient were also included.” In the materials and methods.

3

The authors should report a table with overall patient demographics and clinicopathologic findings (age range, percentage of stage and grade and histology of tumors).

Thanh you for this comment.

We initially wanted to add demographics and clinicopathologic findings to our tables however this objective was not possible because the parameters were not documented clearly in most of the review. In addition, most articles referred to the demographics of their general population and not the MMRd subtype. There were a great lack of information.

However we have specified this point within the limits of our paper.

4

The authors did not carry out association analyses, however, apart from the non-homogeneity of the samples, some associative aspects between the clinical characteristics and the markers taken into consideration should be analyzed through different statistical tests. At least select the studies with which the authors believe they can do a meta-analysis.

The risk is that this manuscript will remain a mere sequence of lists of papers with qualitative conclusions. The scientific level that the journal has reached requires this effort.

In agreement with the editor, we chose to do a review of the literature and not a meta-analysis. In this sense, our methodology was not designed to be a meta-analysis.

If you or the editor wish, we can add this to the discussion of our limitations.

5

  In this recent paper: Jumaah AS, Al-Haddad HS, Salem MM, McAllister KA, Yasseen AA. Mismatch repair deficiency and clinicopathological characteristics in endometrial carcinoma: a systematic review and meta-analysis. J Pathol Transl Med. 2021 May;55(3):202-211. doi: 10.4132/jptm.2021.02.19, the authors conclude that the frequency of MMR deficiency in EC remains elusive. The authors should provide comment on any inconsistencies with their results.

Thank you for this comment.

We added this paragraph in the discussion:

“Our Systematic review confirms Jumaah et al’s meta-analysis which shows a large variation in the diagnosis of MMRds tumors. This depends on the initial study population which may include only low or high grade tumors, lynch families or concurrent cancers. In contrast to us, this meta-analysis included MMRd tumors diagnosed by molecular biology which probably allowed them to obtain a larger variation. From a prognostic point of view, our articles are concordant since they point out that MMRd tumors have a more favorable prognosis compared with MMR-proficient and that their immune context is a key point of immunotherapy [Jumaah].”

Round 2

Reviewer 1 Report

The authors have satisfied all suggestions and the manuscript has gained in quality. However, I recommend a paragraph in the discussion, where the authors express their own ideas about the interpretation of the results and if it suggests future applications with the results described in the study.

Author Response

Reviewer 1

The authors have satisfied all suggestions and the manuscript has gained in quality. However, I recommend a paragraph in the discussion, where the authors express their own ideas about the interpretation of the results and if it suggests future applications with the results described in the study.

Thank you for this comment.

We added this paragraph in the discussion:

This review of the literature allows us to state that in order not to underestimate the MSI tumors in the EC it is strongly recommended to use the 4 proteins of the MMR system by the IHC technique which remains the least expensive and most reproducible. Also, our review of the literature has updated the data in the literature on the prognosis of these tumors. The research on MSI tumors tends towards a better prognosis thanks to their particular immune context which are the key to many immunotherapies.

Reviewer 2 Report

I appreciate the changes and the efforts of the authors and I understand some choices. The manuscript has certainly improved.

Author Response

Reviewer 2

I appreciate the changes and the efforts of the authors and I understand some choices. The manuscript has certainly improved.

Thank you for this comment.
